# Informing the development of diagnostic criteria for differential diagnosis of alcohol-related cognitive impairment (ARCI) among heavy drinkers: A systematic scoping review

**Lisa Jones**[1,2]*, **Lynn Owens**[1,3,4], **Andrew Thompson**[1,4], **Ian Gilmore**[1], **Paul Richardson**[1,3]

1 Liverpool Centre for Alcohol Research, Institute of Translational Medicine, University of Liverpool, Liverpool, United Kingdom, 2 Public Health Institute, Faculty of Health, Liverpool John Moores University, Liverpool, United Kingdom, 3 Department of Gastroenterology and Hepatology, Liverpool University Hospitals NHS Foundation Trust, Liverpool, United Kingdom, 4 Institute of Systems, Molecular and Integrative Biology, University of Liverpool, Liverpool, United Kingdom

* l.jones1@ljmu.ac.uk

**Data Availability Statement:** All data generated or analysed during this study are included within the manuscript and its supporting information files.

## Abstract

### Background

Early detection and diagnosis of alcohol-related cognitive impairment (ARCI) among heavy drinkers is crucial to facilitating appropriate referral and treatment. However, there is lack of consensus in defining diagnostic criteria for ARCI. Uncertainty in attributing a diagnosis of suspected ARCI commonly arises in clinical practice and opportunities to intervene are missed. A systematic scoping review approach was taken to (i) summarise evidence relating to screening or diagnostic criteria used in clinical studies to detect ARCI; and (ii) to determine the extent of the research available about cognitive assessment tools used in 'point-of-care' screening or assessment of patients with suspected non-Korsakoff Syndrome forms of ARCI.

### Methods

We searched Medline, PsycINFO, Cinahl and the Web of Science, screened reference lists and carried out forward and backwards citation searching to identify clinical studies about screening, diagnosis or assessment of patients with suspected ARCI.

### Results

In total, only 7 studies met our primary objective and reported on modifications to existing definitions or diagnostic criteria for ARCI. These studies revealed a lack of coordinated research and progress towards the development and standardisation of diagnostic criteria for ARCI. Cognitive screening tools are commonly used in practice to support a diagnosis of ARCI, and as a secondary objective we included an additional 12 studies, which covered a range of settings and patient populations relevant to screening, diagnosis or assessment in acute, secondary or community 'point-of-care' settings. Across two studies with a defined

**Funding:** This study was supported by charitable funds (ref: A0085/20CF) from the Gastroenterology Fund of the Liverpool University Hospitals NHS Foundation Trust (formerly, the Royal Liverpool and Broadgreen University Hospitals NHS Trust). The funders had no role in study design, data collection and analysis, decision to publish, or preparation of the manuscript.

**Competing interests:** No. The authors have declared that no competing interests exist.

ARCI patient sample and a further four studies with an alcohol use disorder patient sample, the accuracy, validity and/or reliability of seven different cognitive assessment tools were examined. The remaining seven studies reported descriptive findings, demonstrating the lack of evidence available to draw conclusions about which tools are most appropriate for screening patients with suspected ARCI.

## Conclusion

This review confirms the scarcity of evidence available on the screening, diagnosis or assessment of patients with suspected ARCI. The lack of evidence is an important barrier to the development of clear guidelines for diagnosing ARCI, which would ultimately improve the real-world management and treatment of patients with ARCI.

## Introduction

Chronic excessive alcohol consumption has long been recognised as a cause of cognitive impairment, although understanding of the underlying neurobiology is limited [1–4]. Nevertheless, several neurodegenerative changes are thought to be caused by heavy drinking. Indeed, patients with an Alcohol Use Disorder (AUD) often present with a variety of cognitive deficits and this leads to difficulties in determining the cause, severity and duration of symptoms because features are present in other causes of brain injury such as Wernicke's or hepatic encephalopathy [5]. Reflecting etiological and nosological debates in the field [6], variant terms have come into use as umbrella terms to characterise cognitive impairment in heavy drinkers [7, 8], including "alcohol-related brain damage", "alcohol-related dementia", and "alcoholic amnesia syndrome" [9]. A further source of variation has arisen in how these umbrella terms have been used by clinicians and researchers, both to encompass and to provide a distinction from the more severe Wernicke-Korsakoff's Syndrome (WKS) [10]. For the purposes of this scoping review and reflecting the variant terms in use, we have applied the term ARCI it in its broadest sense. We have included articles following either application of the terminology, for example, by recognising that alcohol-related dementia and ARCI may be used to refer to the same suspected condition, and that umbrella terms such as alcohol-related brain damage may either encompass or exclude WKS.

The diagnostic landscape and associated nomenclature have hindered clarity and certainty in both clinical practice and research studies. Relevant to this context is that the diagnostic criteria within the two major classification systems, the International Classification of Disease (ICD) manual and the Diagnostic Statistical Manual (DSM), have previously focused on the two main syndromes of WKS and ARD, rather than providing unified criteria [7]. For example, within the ICD-10 [11] (and ICD-11, which came into effect from 1st January 2022) a diagnostic distinction is made between amnestic disorder (WKS) and alcohol-induced dementia. However, ARD has not consistently been recognised as a discrete clinical entity [12]. Further, based on criticisms directed at the DSM-IV criteria for alcohol-related dementia, Oslin et al. [13] proposed alternative diagnostic criteria in 1998 (Table 1). These criteria were later adapted by Wilson et al. [14] into a 'probable diagnosis of alcohol-related brain damage' and have, to an extent, been adopted into UK clinical practice [15]. More recently the DSM-5 [16] has adopted the term "alcohol-related neurocognitive disorders", distinguishing both between the type (non-amnestic-type versus confabulating-amnestic type) and severity (major or minor) of the impairment. This new classification is therefore more aligned with the current,

**Table 1. Classification of 'probable' ARD (reproduced from [13]).**

*Probable* Alcohol Related Dementia
A. The criteria for the clinical diagnosis of Probable Alcohol Related Dementia include the following:
 1. A clinical diagnosis of dementia at least 60 days after the last exposure to alcohol.
 2. Significant alcohol use as defined by a minimum average of 35 standard drinks per week for men (28 for women) for greater than a period of 5 years. The period of significant alcohol use must occur within 3 years of the initial onset of Dementia.
B. The diagnosis of Alcohol Related Dementia is supported by the presence of any of the following:
 1. Alcohol related hepatic, pancreatic, gastrointestinal, cardiovascular, or renal disease i.e. other end-organ damage.
 2. Ataxia or peripheral sensory polyneuropathy (not attributable to other specific causes).
 3. Beyond 60 days of abstinence, the cognitive impairment stabilizes or improves.
 4. After 60 days of abstinence, any neuroimaging evidence of ventricular or sulcal dilatation improves.
 5. Neuroimaging evidence of cerebellar atrophy, especially of the vermis.
C. The following clinical features cast doubt on the diagnosis of Alcohol Related Dementia.
 1. The presence of language impairment, especially dysnomia or anomia.
 2. The presence of focal neurologic signs or symptoms (except ataxia or peripheral sensory polyneuropathy).
 3. Neuroimaging evidence for cortical or subcortical infarction, subdural hematoma, or other focal brain pathology.
 4. Elevated Hachinski Ischemia Scale score.
D. Clinical features that are neither supportive nor cast doubt on the diagnosis of Alcohol Related Dementia included:
 1. Neuroimaging evidence of cortical atrophy.
 2. The presence of periventricular or deep white matter lesions on neuroimaging in the absence of focal infarct(s).
 3. The presence of the Apolipoprotein e4 allele.

prevailing concept of ARCI as being on a continuum [5, 15, 17, 18]. Key issues remain, however, with more rigorous empirical research needed to validate both new and existing criteria for ARCI among heavy drinkers in clinical practice.

Detection and diagnosis of ARCI at the earliest opportunity is crucial to facilitating appropriate referral and treatment [6, 19]. The major features of presentation in this patient group relate to cognitive impairments that affect memory and executive functioning with subsequent behavioural change, features known to adversely affect outcomes [20]. People with ARCI may experience variable psychiatric and social problems as a result of these impairments including difficulties with reasoning and problems with impulse control [14]. This may go some way to explaining why they have difficulties with adherence to traditional approaches to alcohol treatment, leading to development of multiple co-morbidities and poor treatment outcomes overall [21].

Evidence from the broader spectrum of conditions associated with cognitive impairment suggests that recognition and a diagnosis of ARCI would significantly improve the long-term prognosis for patients. Among patients with Korsakoff's Syndrome, Smith and Hillman [22] estimated that with treatment, full recovery can be achieved in approximately 25%, and among the remainder, 50% can achieve a partial or minor recovery. However, there are currently significant barriers to the timely recognition and diagnosis of ARCI including a lack of training and clear guidance for clinicians, stigma towards the patient group and fragmentation of care [23], resulting in a disproportionate burden being placed on health services [24]. This underlines the importance of the need for the rapid identification of patients who present within acute hospital or community settings with an AUD and cognitive impairment. Extensive neuropsychological assessment is the gold standard for identifying cognitive impairments among patients with a history of heavy drinking. Unfortunately, most acute and community setting do not have timely access to the expertise needed to perform these assessments, and it follows that this approach is not feasible outside of specialist psychiatric or neuropsychiatric settings. Brief cognitive assessment tools, such as the Montreal Cognitive Assessment (MoCA) [25] have therefore been used in combination with other criteria to identify patients with suspected ARCI in non-specialist settings. Heirene et al. [10] recently published a systematic

review that examined evidence for neuropsychological tests used to assess ARCI, including brief cognitive screening instruments. However, the majority of the studies included in the review investigated WKS and studies on non-WKS forms of ARCI were lacking.

Without a consensus on the best diagnostic criteria for suspected ARBI, a lack of clarity and uncertainty in attributing a diagnosis will continue, and opportunities to intervene will be missed. The aim of this scoping review was two-fold. Our primary objective was to systematically map the literature and summarise evidence relating to screening or diagnostic criteria used in clinical studies to detect ARCI and explore how criteria have been adapted or validated over time. In practice, few studies met these criteria, and so a secondary objective was established to review the extent of the literature available on cognitive assessment tools used in 'point-of-care' screening or assessment of patients with suspected non-WKS forms of ARCI.

## Materials and methods

The review was carried out in accordance with the Preferred Reporting Items for Systematic reviews and Meta-Analyses extension for Scoping Reviews (PRISMA-ScR) and a checklist is presented in the supporting information. A scoping review protocol was developed in advance (but not prospectively registered) and is provided in the S1 Protocol. We included papers reporting on the screening, diagnosis, or assessment of people with suspected ARCI.

### Inclusion criteria

To meet our primary objective, we included articles that reported on modifications to existing definitions or diagnostic criteria or that presented new diagnostic criteria (including original research and consensus/statement articles), as well as those that referred to, discussed, or compared existing definitions of ARCI as per our protocol. Original, primary research and consensus/statement articles published in English since 1990 were eligible for inclusion as Oslin et al. [13] proposed the first diagnostic criteria for alcohol-related dementia in 1998 with the purpose of stimulating further research (Table 1). We planned to exclude studies that solely used the ICD-10 or DSM-IV criteria, on the basis that these diagnostic systems don't provide unifying criteria for ARCI. Articles solely about the treatment and management of ARCI were excluded as were studies that solely assessed WKS or Wernicke encephalopathy.

To meet our secondary objective, we selected studies relevant to the screening, diagnosis, or assessment of patients with suspected ARCI, or cognitive impairment among patients with an alcohol use disorder, in acute, secondary or community 'point-of-care' settings. Studies that described screening tools with the following characteristics were included: (i) designed to screen for cognitive impairment, (ii) duration of the tool was described as brief or could be administered in 30 minutes or less, (iii) tool administered directly to patients, and (iv) patients selected for inclusion in accordance with diagnostic criteria for either ARCI (e.g. studies referring to 'alcohol-related brain damage' or 'alcohol-related brain injury') or DSM (or other recognised) criteria for alcohol dependence.

### Searching for and selecting relevant studies

A database of English-language articles was compiled in EndNote based on systematic searches of the literature. We developed an initial targeted search strategy for key index papers in Medline and PsycINFO, by combining keyword terms for alcohol and brain injury, with terms for screening, diagnosis, and assessment. These initial searches were conducted in October 2019 and rapidly screened by a single reviewer. A revised search strategy, which focused on a narrower set of terms for ARCI without the diagnosis terms from the original framework, was subsequently developed (supporting information). This search strategy is provided in the S1

Table. Searches were conducted in Medline (Ovid), PsycINFO (Proquest), Cinahl (EBSCO-host), Social Citation Index and Social Sciences Citation Index (Web of Knowledge) in December 2019. The searches were updated in September 2020 and February 2021 and finally in October 2021.We also sought relevant literature through supplementary searches of Google Scholar for grey literature, relevant websites (e.g. Alcohol Change UK, Public Health England, Scottish Executive), manual screening of reference lists, and forwards and backwards citation searching in Scopus.

We screened the results of the searches to identify relevant studies in two stages. Firstly, two reviewers (LJ & AA) double screened 20% of the titles and abstracts identified for potential inclusion. Interrater reliability was not recorded as a level of disagreement was anticipated based on the wide use of terminology in the field. Discrepancies in study selection were discussed and resolved and used to develop confidence in the screening of the remaining titles and abstracts, which was carried out by a single reviewer (LJ). Next, full-text publications of any potentially relevant titles were obtained and assessed against the inclusion criteria. Three reviewers (LJ, LO & AT) independently screened all potentially relevant articles, and decisions on inclusion were reached through consensus.

### Data charting process and quality assessment

We used a coding strategy to concisely record and chart the information from the included literature (S1 Appendix). This included details about study type and setting (e.g. community or hospital), the eligibility criteria for participants, methods used to screen, diagnose, or assess patients, and brief details of the study findings. Data charting was done independently by one reviewer (LJ). For studies that reported on the accuracy, validity and/or reliability of a cognitive assessment tool test validity, a quality assessment tool developed by Heirene et al. [10] was used to assess the quality of these studies. Quality assessment was done independently by one reviewer (LJ) (S2 Table).

### Data synthesis

Studies were grouped according to which of the objectives they addressed; whether they (i) presented new or adapted screening or diagnostic criteria for ARCI (primary objective); (ii) evaluated the accuracy, validity and/or reliability of a cognitive assessment tool (secondary objective); or (iii) presented a descriptive summary of a population based on cognitive assessment of patients with suspected ARCI (secondary objective). The limited availability of outcome data meant that the data are reported as a descriptive summary of the evidence.

## Results

In total, 3,568 unique records were identified up to October 2021 and screened for inclusion in the review following the removal of foreign language articles (Fig 1).

A total of 139 studies were excluded at the full text screening stage (Fig 1). Twelve studies were considered 'near misses' and judged to have narrowly failed to meet the inclusion criteria in respect of the review's secondary objective as the samples included patients with diagnoses of dependence on substances other than alcohol. We chose to exclude those studies post hoc, where the sample included patients with a primary diagnosis (based on DSM or other recognised criteria) of abuse or dependence on a substance other than alcohol.

Nineteen records were kept and comprised the final set of included studies in the review. Three studies [14, 26, 27] contributed to both scoping review objectives. Eight studies [9, 14, 26–31] included participants with alcohol-related dementia or ARCI and were included in the summary of evidence addressing our primary objective relating to new or adapted diagnostic

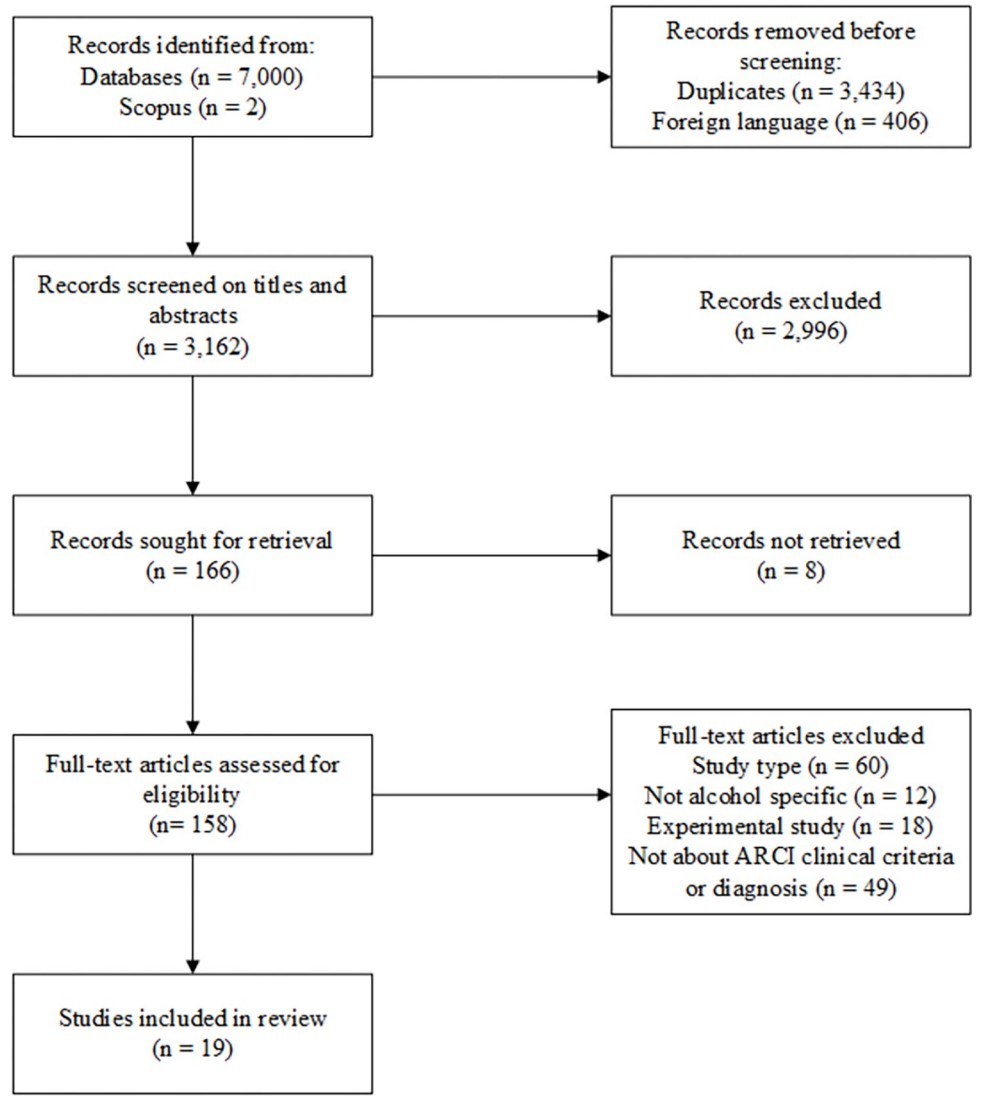

**Fig 1. Study inclusion flow diagram.**

or screening criteria for ARCI. Twelve studies [14, 26, 27, 32–42] were included in the summary of evidence addressing our secondary objective relating to cognitive assessment tools. Seven studies [26, 27, 34, 35, 37, 38, 41] evaluated the accuracy, validity and/or reliability of a cognitive assessment tool; two studies [26, 27] included patients with a diagnosis of ARCI and five studies [34, 35, 37, 38, 41] with patients with a diagnosis of AUD. Seven further studies [14, 32, 33, 36, 39, 40, 42] were descriptive reports about the use of cognitive assessment tools in screening and/or assessment. Two studies [14, 36] included patients with a diagnosis of 'alcohol-related brain damage' and five studies [32, 33, 39, 40, 42] included samples with an AUD diagnosis.

## New or adapted screening or diagnostic criteria for ARCI

Diagnostic/screening criteria are summarised for eight studies [9, 14, 26–31] in Table 2. Four studies were from the UK [9, 14, 26, 29], two from the USA [30, 31], and one each from Canada [28] and the Netherlands [27]. Provisional diagnostic criteria for 'probable' alcohol-related dementia were published by Oslin et al. in 1998 [13]. The study by Carlen et al. [28] preceded

**Table 2. Summary of reported diagnostic criteria for ARCI.**

| Author, Year | Country | Diagnosis [a] | Setting | Diagnostic/screening criteria | | |
|---|---|---|---|---|---|---|
| | | | | **Cognitive impairment** | **Alcohol-related** | **Other** |
| Carlen et al., 1994 [28] | Canada | ARD | Long-term care facilities | MMSE score <24 OR performance scores >2 SD below control means on CERAD screening tests in at least two areas of function (language, memory, or praxis). | History of alcohol use for >5-year period and average >6 oz/day. | Lack of progression in cognitive decline for at least the 1st year subsequent to institutionalization; exclusion of other causes of dementia. |
| Oslin & Cary, 2003 [30] (validation of Oslin et al., 1998) | USA | Probable ARD | Nursing Home | Clinical diagnosis of dementia at least 60 days after the last exposure to alcohol. | Minimum average of 35 standard drinks per week for men (28 for women) for >5 years and within 3 years of the initial onset of dementia. | Also specifies additional criteria supporting a diagnosis of ARD and criteria which may cast doubt. |
| Schmidt et al., 2005 [31] (adapted from Oslin et al., 1998) | USA | ARD | Memory assessment programme | Diagnosis of dementia at least 60 days after last exposure to alcohol. | 35 alcoholic drinks per week for men (28 for women) for >5-year period. | No history of an acute onset of symptoms associated with WE; lack of focal neurological signs (except ataxia or peripheral sensory polyneuropathy). |
| Gilchrist & Morrison, 2005 [29] | UK | ARBD | Homeless hostel | ACE score <88. | Hazardous drinking in the last year identified by FAST score ≥3; current alcohol dependence based on LDQ last week score ≥9 OR LDQ lifetime score ≥9. | |
| Wilson et al., 2012 [14] (adapted from Oslin & Cary, 2003) | UK | ARBD | Tertiary service for patients with severe ARBI | Confusion, memory problems, doubt about capacity and concerns about risk on discharge, after withdrawal/ physical stabilization. | Probable history of heavy, long-standing alcohol drinking: ≥35 units/week for ≥5 years; 3+ hospital admissions &/or A&E in 1 year directly or indirectly with alcohol ingestion OR 1 + delayed hospital discharges in last 12 months. | |
| Wester et al., 2013 [27] | The Netherlands | Non-WKS forms of ARCI | Clinic for patients with suspected cognitive impairments due to alcohol-use disorder | Suspected cognitive impairments due to alcohol-use disorder; no severe memory deficits. | DSM-IV-TR criteria for alcohol dependence. | Not fulfilling the criteria for KS. |
| Brown et al., 2019 [26] (based on Wilson et al., 2012) | UK | ARBD | Glasgow specialist ARBD service | Evidence of cognitive deficits typically associated with alcohol-dependence. | Chronic and excessive alcohol history (not defined). | Neuroimaging evidence of structural brain change; psychosocial deterioration. |
| Thompson et al., 2020 [5] (based on Wilson et al., 2012) | UK | ARBI | Acute hospital | MoCA score ≤23. | 3+ alcohol-related admissions in 1 year; OR 2 alcohol-related admissions in any given 30-day period; OR patient/ significant other had concerns regarding cognition. | |

[a] Based on terms used by the authors of the studies.

ARBD, alcohol-related brain damage; ARBI, alcohol-related brain injury; ARCI, alcohol-related cognitive impairments; ARD, alcohol-related dementia; CERAD, Consortium to Establish a Registry for Alzheimer's Disease; FAST, Fast Alcohol Screening Test; KS, Korsakoff's Syndrome; LDQ, = Leeds Dependence Questionnaire; MMSE, Mini Mental State Examination; MoCA, Montreal Cognitive Assessment; WE, Wernicke's encephalopathy.

these criteria but are largely similar, for example, with respect to history of chronic and excessive alcohol use. Oslin & Cary [30] sought to validate the 1998 criteria through a longitudinal study which explored the functional and cognitive course of alcohol-related dementia, and two further studies [14, 31] reported being based on adapted Oslin 1998/Oslin & Cary 2003 criteria [13, 30]. Schmidt et al. [31] incorporated additional criteria for the diagnosis of alcohol-related dementia, specifying the exclusion of patients with a history of an acute onset of symptoms associated with WE. Wilson et al. [14] reported screening criteria for referral to a specialist service. The criteria they developed were designed to be used by non-trained generic nurses in an acute hospital setting and included additional items relating to hospital admissions and delayed discharges. In keeping with their design for non-trained medical staff, they don't state that a clinical diagnosis of dementia is required, referring instead to "confusion, memory problems" and "doubt about capacity".

Two studies [9, 29] incorporated screening/diagnostic cut-offs on a cognitive screening tool as part of screening processes for ARCI. Gilchrist & Morris [29] explored the prevalence of ARCI among homeless people living in five large hostels in Glasgow. The screening criteria for ARCI was based on an assessment of cognitive impairment with the Addenbrooke's Cognitive Examination (ACE; scoring less than 88) and meeting criteria for hazardous drinking and current or lifetime alcohol dependence (Table 3). The study by Thompson et al. [9] was set in an acute hospital. The screening criteria for ARCI were based on Wilson et al. [14] but also incorporated an assessment with MoCA, with a score of 23 or less considered evidence of ARCI. Two studies [26, 27] reported on ARCI patients sampled from clinics or service with a specialist remit for treating ARCI. Wester et al. [27] established a group of patients with non-KS forms of ARCI based on fulfilling the DSM-IV-TR criteria for alcohol dependence but without severe memory deficits (the criteria for Korsakoff Syndrome). Brown et al. [26] included participants from a specialist service but the criteria for the diagnoses was unclear. However, Brown et al. [26] reported that diagnoses were based on 'most or all' of a set of criteria that were in line with those of Wilson et al. [14].

## Cognitive assessment tools used in point-of-care screening or assessment

A definitive diagnosis of cognitive impairment among patients with a history of heavy drinking is achieved through extensive neuropsychological assessment, but this approach is not feasible outside of specialist psychiatric or neuropsychiatric settings. Therefore, brief cognitive screening tools are commonly used in practice to support a diagnosis of suspected ARCI. A summary of the nine different cognitive screening tests examined or described across the included studies is presented in Table 3. The most commonly used tool across the included studies was the MoCA. Across the nine studies that included patients with ARCI [9, 14, 26–31, 36], tools used were the Mini-Mental State Examination (MMSE), Addenbrooke's Cognitive Examination (ACE), MoCA, Cambridge Neuropsychological Test Automated Battery (CANTAB) and the Repeatable Battery for the Assessment of Neuropsychological Status (RBANS).

## Studies about the accuracy, validity and/or reliability of a cognitive assessment tool

Seven studies [26, 27, 34, 35, 37, 38, 41] examined the accuracy, validity and/or reliability of a cognitive assessment tool (Table 4). The results of the full quality assessment is provided in the S2 Table). Heirene et al. [10] draw attention to several individual level factors that may impact on cognitive test outcomes including, illicit substance use, comorbid psychopathology, psychiatric medication use, and cerebrovascular disease and traumatic brain injuries. An abstinence period from illicit substances, of at least 6 weeks is typically needed before reliable neurological

**Table 3. Summary of cognitive screening tools.**

| Assessment or Screening Test | Summary | Tasks & functions assessed | Admin. / scoring time | Max score | Reference(s) |
|---|---|---|---|---|---|
| ACE, ACE-R, ACE-III | Brief cognitive test battery. ACE and ACE-R versions incorporated the MMSE. | Orientation, Attention, Memory, Verbal fluency, Language, Visuospatial | 15–20 mins | 100 | Brown et al., 2019* [26]; Gilchrist & Morrison, 2005* [29]; Wilson et al., 2012* [14]; Rao, 2016 [39] |
| BEARNI | Brief assessment giving five sub-scores | Episodic memory, Working memory, flexibility, visuospatial, ataxia | 15–20 mins | 30; cognitive (no ataxia) 22 | Pelletier et al., 2018 [38]; Ritz et al., 2015 [41] |
| CANTAB | Computerised assessment comprising 7 subtests. | Episodic memory, sustained visual attention, spatial planning, working memory, rule acquisition | 3 mins for screening, 3–10 mins for primary tests | Unclear | Horton et al., 2015* [36] |
| MMSE | Brief cognitive assessment. Subtest & total score | Spatial and temporal orientation, Attention Calculation, Language, Memory, Comprehension, Copy design | 5–10 mins | 30, standard cut-off <24 | Oslin & Cary, 2003* [30]; Reid et al., 2002 [40]; Carlen et al., 1994* [28]; Schmidt et al., 2005* [31] |
| MoCA | Brief cognitive screening measure. 14 tasks. | Memory, Executive function, Attention & Concentration, Language, Visuospatial, Orientation | | 30, standard cut-off <26 | Thompson et al., 2020* [5]; Wester et al., 2013* [27]; Alarcon et al., 2015 [32]; Ewert et al., 2018 [35]; Pelletier et al., 2018 [38] |
| MST | Short screening instrument for cognitive impairment. Consists of three recall task: word recall, sentence recall and figure recall. | Not reported | Not reported | Not reported | Taylor et al., 1997 [42] |
| NIS | 50-item self-report scale with eight sub-scales: Global Measure of Impairment, Total Items Checked, Symptom Intensity Measure, Lie scale, General scale, Pathognomic scale, Learning-Verbal scale, and Frustration scale. | Attention, memory, language | Unclear | Optimal NIS scale score defined as impaired T-score >60. | Errico et al., 1990 [34] |
| RBANS | 12 subtests & overall performance score. Converted to age-adjusted norms scores. Does not provide a targeted assessment of executive functioning. | Visuospatial/ constructional, Language, Attention, Memory | 20–30 mins | Mean 100 (sd 15), standard cut-off <88 & <83 | Brown et al., 2019* [26]; Cao et al., 2021 [33] |
| TEDCA | Developed as a screening test specifically for patients with a history of alcoholism. Test/items used to construct the tool: ROCF, Bender Test, Direct Digits, Inverse Digits, Numbers and Letters, Learning List, TMT-B, Similarities, Go-No Go. | Visuospatial cognition, memory/ learning, executive function | 8–10 mins | Unclear | Jurado-Barba et al., 2017 [37] |

*ARCI patient sample

ACE, Addenbrooke's Cognitive Examination; BEARNI, Brief Evaluation of Alcohol-Related Neuropsychological Impairments; CANTAB, Cambridge Neuropsychological Test Automated Battery; MMSE, Mini-Mental State Examination; MoCA, Montreal Cognitive Assessment; NIS, Neuropsychological Impairment Scale; RBANS, Repeatable Battery for the Assessment of Neuropsychological Status; ROCF, Rey–Osterrieth Complex Figure; TEDCA, TEst of Detection of Cognitive impairment in Alcoholism; TMT-B, Trail Making Test Part B.

assessment can be carried out [43]. Periods of abstinence prior to assessment varied across the seven studies, with only one study [26] reporting a period close to the minimum 6 weeks. All seven studies provided adequate descriptions of the included patient samples, but exclusions based on comorbid or confounding conditions were not consistent across the studies. For two studies [27, 34], exclusions were not clear or not reported. Three studies excluded patients with comorbid psychopathology [35, 37, 38] and two studies referred to the exclusion of patients with brain injuries [26, 37]. Effect sizes were only reported in two of six studies where

**Table 4. Summary of studies: Accuracy, validity and/or reliability of a cognitive assessment tool.**

| Reference | Country | Setting | Participants | Assessment(s) Period of abstinence | Main findings |
|---|---|---|---|---|---|
| Brown et al., 2019 [26] | UK | Glasgow specialist ARBD service and others (for AUD) | 28 ARBD based on Wilson 2012 criteria (11 KS, 17 other) | ACE-III; RBANS Abstinent from alcohol and other substances for min. 5 weeks. | Both tests differentiated between AUD and ARBI groups. Findings considered to support the use of both tests in clinical assessments of alcohol users. |
| Wester et al., 2013 [27] | The Netherlands | Clinic for patients with suspected cognitive impairments due to alcohol-use disorder | 26 ARCI (defined as non-KS); AUD based on DSM-IV-TR but did not meet criteria for KS. (20 KS; 33 controls) | MoCA; RBMT-3. MoCA administered at intake RBMT-3 administered 6–8 weeks after admission | MoCA was able to discriminate mild and more severe forms of memory impairment. Moderate discriminatory power to distinguish between WKS and non-WKS forms of ARBI. |
| Errico et al., 1990 [34] | USA | Veterans Administration Medical Center | 73 AUD (254 controls) | NIS Administered 7 days after admission to treatment programme | Higher NIS scale scores associated with poorer the performance on the NP tests (association 'only modest') |
| Ewert et al., 2018 [35] | France | Hospital-based substance use disorder rehabilitation centre | 56 AUD (cognitive impairment, 31; no cognitive impairment, 25) | MoCA Administered within 2 weeks and at least 7 days after alcohol withdrawal | NP tests were significantly correlated with the MoCA and more than 80% of AUD patients with one or more cognitive deficits were classified correctly using either corrected or uncorrected MoCA scores. |
| Jurado-Barba et al., 2017 [37] | Spain | Hospital-based Addictive Behaviour Unit | 90 patients with an AUD (DSM-5) | TEDCA Not reported | Test demonstrated reliability and good diagnostic validity |
| Pelletier et al., 2018 [38] | France | Rehabilitation centre | 90 AUD | MoCA, BEARNI Administered about 7–10 days after alcohol withdrawal | Cognitive domains assessed matched well with gold standard NP tests. BEARNI showed poor sensitivity. PPV for MoCA significantly better than BEARNI. |
| Ritz et al., 2015 [41] | France | Hospital-based, receiving withdrawal treatment | 73 AUD (58 controls) | BEARNI Administered immediately after withdrawal | Specificity and PPV showed test had high sensitivity for mild and moderate-to-severe impairment. Specificity was poor. |

ACE, Addenbrooke's Cognitive Examination; AUD, Alcohol Use Disorder; BEARNI, Brief Evaluation of Alcohol-Related Neuropsychological Impairments; CANTAB, Cambridge Neuropsychological Test Automated Battery; KS, Korsakoff Syndrome; MMSE, Mini-Mental State Examination; MoCA, Montreal Cognitive Assessment; NIS, Neuropsychological Impairment Scale; NP, neuropsychological; RBANS, Repeatable Battery for the Assessment of Neuropsychological Status; RBMT, Rivermead Behavioral Memory Test; TEDCA, TEst of Detection of Cognitive impairment in Alcoholism.

this was deemed appropriate [26, 38]. As shown in Table 4, studies typically included small to moderate sample sizes. Four studies [26, 27, 35, 41] reported clearly that they had adjusted for the risk of type 1 error in their analyses.

**ARCI patients.** Two studies examined test validity with ARCI samples [26, 27]. Brown et al. [26] examined the suitability of the third edition of the ACE (ACE-III) and the RBANS. Both the ACE-III and RBANS distinguished between the AUD and ARCI patient groups. Wester et al. [27] examined the discriminatory power of the MoCA screening tool finding that it had moderate discriminatory power to distinguish between WKS and non-WKS forms of ARCI. However, neither study included a comparison with a reference standard.

**AUD patients.** Five studies [34, 35, 37, 38, 41] examined the validity of a range of cognitive assessment tools among AUD patients with compared to the reference standard of a more extensive neuropsychological assessment. Higher scores on the Neuropsychological Impairment Scale (NIS; a self-report scale) [34] were associated with poorer the performance on the reference standard neuropsychological tests, but this association was noted to be 'only modest'. The MoCA was examined in two studies [35, 38]. Ewert et al. [35] found that the reference standard neuropsychological tests were significantly correlated with the MoCA and

that more than 80% of AUD patients with one or more cognitive deficits were classified correctly using either corrected or uncorrected MoCA scores. Pelletier 2018 [38] compared the performance of another cognitive assessment tool, BEARNI with MoCA. In both tests, the cognitive domains matched well with the reference standard neuropsychological tests. However, BEARNI showed poor sensitivity. The psychometric properties of BEARNI were also examined by Ritz et al. [41]. Based on specificity and the positive predictive value, the test had high sensitivity for both mild and moderate-to-severe impairment, but its specificity was poor. Jurado-Barba et al. [37] reported on a two-phase process to select items to compose the TEDCA (Test of detection of cognitive impairment in alcoholism). In phase 2 tests, TEDCA was used to assess 90 patients with an AUD (DSM-V) and demonstrated reliability and good diagnostic validity.

## Descriptive reports about cognitive assessment tools

Six studies [32, 33, 36, 39, 40, 42] provided descriptive reports about a cognitive assessment tool (Table 5).

**ARCI patients.** Two studies [14, 36] explored cognitive functioning among ARCI patients admitted to specialist tertiary care settings. Patients in the study by Wilson et al. [14], underwent assessment at admission to the service with a revised version of the ACE (ACE-R). The group average score for 22 patients was 65.7 (range 30–93), indicating a considerable range of

**Table 5. Summary of studies: Descriptive reports of a cognitive assessment tool.**

| Reference | Country | Setting | Participants | Assessment(s) Period of abstinence | Main findings |
|---|---|---|---|---|---|
| Alarcon et al., 2015 [32] | France | Hospital-based addiction treatment unit | 166 AUD (DSM-IV) | MoCA Participants abstinent for 1–2 weeks | High rate of cognitive deficits identified. Visuospatial construction, fluency, abstraction and delayed recall were shown to be impaired, while naming, attention, orientation and repeat (language domain) were not. |
| Cao et al., 2021 [33] | China | Hospital-based, psychiatric department | 60 male AUD (40 controls) | RBANS Participants abstinent for 7 days | Speech function, attention function, delayed memory and immediate memory significantly reduced compared to control. No difference in visual breadth. |
| Horton et al., 2015 [36] | UK | Abstinence-based 'ARBD residential rehabilitation service' | 20 ARBD patients | CANTAB Period of abstinence not reported | Participants performed below the normative average on all 5 subtests: episodic memory, sustained visual attention, spatial planning, working memory and rule acquisition and attentional set shifting. |
| Rao, 2016 [39] | UK | Four community mental health teams' caseloads | 25 older patients with ICD-10 diagnosis of alcohol dependence | ACE-III Period of abstinence not reported | Scores based on pre-defined cut-off suggest deficits in attention/orientation, fluency, visuospatial function and memory (68–84% scoring below cut-off). Only 45% scored below cut-off for language. |
| Reid et al., 2002 [40] | USA | Hospital-based geriatric assessment centre | 801 older patients | MMSE Period of abstinence not reported | No difference in total scores across alcohol exposure categories or on history of alcohol abuse or dependence for patients with MMSE <24. |
| Taylor et al., 1997 [42] | Australia | Detoxification unit | 16 AUD (3 had ARBI diagnosis) (16 controls) | Memory Screening Test Period of abstinence not reported | AUD patients made a greater number of errors on each recall task than controls. No difference in mean number of errors between groups on individual task. Group mean for total errors was higher in AUD patients |
| Wilson et al., 2012 [14] | UK | Tertiary service for patients with severe ARBI | 41 ARBI patients | ACE-R Period of abstinence unclear | Group average ACE-R score indicated a considerable range of cognitive impairment. |

ACE, Addenbrooke's Cognitive Examination; ARBD, alcohol-related brain damage; ARBI, alcohol-related brain injury; AUD, Alcohol Use Disorder; CANTAB, Cambridge Neuropsychological Test Automated Battery; MMSE, Mini-Mental State Examination; MoCA, Montreal Cognitive Assessment; RBANS, Repeatable Battery for the Assessment of Neuropsychological Status.

cognitive impairment. Horton et al. [36] investigated the neurocognitive, psychosocial and everyday functioning of a group of individuals with ARCI. Neurocognitive assessment was undertaken with CANTAB and participants were found to have performed below the normative average on five subtests. The findings showed compromised performance in the domains of memory, attention and executive functioning, which the authors suggest are indicative of episodic memory impairments.

**AUD patients.** Five further studies [32, 33, 39, 40, 42] were descriptive reports of cognitive functioning among AUD patients. A range of different cognitive assessment tools were used across the included studies. Settings included hospital-based treatment units for alcohol dependence [32, 33] and a detoxification unit [42]. Based on assessment with MoCA, Alarcon et al. [32] identified a high rate of cognitive deficits in a sample of 166 AUD patients. Visuospatial construction, fluency, abstraction and delayed recall were shown to be impaired, while naming, attention, orientation and repeat (language domain) were not. Cao et al. [33] also found that based on assessment with RBANS, alcohol dependent patients had impairments in immediate memory, attention function, delayed memory and speech function compared to controls. Taylor et al. [42] administered the Memory Screening Test to 16 participants recruited from a detoxification unit. Compared to controls, hazardous drinking participants made a greater number of errors on the Memory Screening Test recall task.

Two studies focused on older adults aged 65 years and older in hospital-based [40] and community [39] settings, respectively. Reid et al. [40] assessed cognitive function with the Mini Mental State Examination but found that increased current alcohol consumption or a lifetime history of alcohol abuse and/or dependence was not associated with characteristic cognitive differences among older patients. Rao [39] carried out a case note analysis of 25 patients under the care of a community mental health team. Using normative data for ACE-III scores in older people, they study found that 76% of AUD patients scored below the cut-off score of 82 with deficits indicated in attention/orientation, fluency, visuospatial function and memory (68–84% scoring below cut-off).

## Discussion

This systematic scoping review identified 19 papers reporting on the screening, diagnosis or assessment of patients with suspected ARCI. Due to limitations in the number of studies identified that reported on modifications to existing definitions or diagnostic criteria for ARCI, or that presented new diagnostic criteria, the scope of this review was broadened. The included studies covered a range of settings and patient populations relevant to screening, diagnosis, or assessment of suspected ARCI in acute, secondary or community 'point-of-care' settings. Details about diagnostic or screening criteria for ARD or ARCI were available across seven studies. Two studies, Schmidt et al. [31] and Wilson et al. [14], reported using adaptions of the Oslin et al. [13] and Oslin and Cary [30] criteria. Schmidt et al. [31] incorporated additional criteria so as to distinguish ARD cases from those meeting criteria for WKS and Wilson et al. [14] included additional items relating to hospital admissions and delayed discharges. Two further studies, both from the UK [9, 26], were then largely based on the Wilson et al. [14] criteria. Although the included studies reported information about the diagnostic or screening criteria used to identify suspected ARCI, except for Oslin and Cary [30], they did not have a primary aim of exploring modifications to existing criteria or present new criteria. There was a lack of discussion across the studies about how and why adaptations were made, and this limits our understanding of any emerging themes or issues that could have been gathered in the context of carrying out this review. Our review therefore shows that although there have been many calls for the diagnostic criteria for ARCI to adequately tested [12, 44] there remains a

lack of coordinated research and progress towards the development and standardisation of diagnostic criteria for ARCI.

According to a recent study carried out in South Wales, UK [15], cognitive screening tools are commonly used in practice to support a diagnosis of ARCI. Only two of the studies identified for inclusion in our review, Gilchrist & Morris [29] and Thompson et al. [9], however, reported incorporating a diagnostic cut-off on a brief cognitive screening tool as part of screening processes for ARCI. Further, we only identified two studies that examined test validity with a defined ARCI patient sample [26, 27]. Broadening the criteria out to patients with AUD, five further studies that examined test validity met the inclusion criteria for our review [34, 35, 37, 38, 41]. Two recent systematic reviews by Heirene et al. [10] and Ko et al. [45], respectively, provide a comprehensive picture of the current evidence available for the use of cognitive screening tools with populations relevant to those with suspected ARCI. MoCA is the best evidenced tool, but methodological limitations across studies demonstrates a need for further research and better translation of evidence to inform real-world clinical practice. There is a clear need for further research efforts to explore whether and how cognitive screening tools should form part of the assessment for suspected ARBI and which of the available tools are most appropriate. With no one tool currently validated, further research on the clinical applicability of the available screening tools is required. Studies should provide information about both the clinical utility (including ease of use and administration time) and psychometric properties (sensitivity and specificity). As shown in this review, although a range of tools have been used to screen for cognitive impairments with ARCI and AUD patient samples, few studies are available on these tools' psychometric properties. MoCA was the only tool for which we identified studies that examined test validity in both an ARCI and an AUD patient sample. As highlighted by Heirene et al. [10], another important area is decisions about when tools should be administered. Neuropsychological functioning improves with the duration of abstinence [43] and for patients at-risk of ARCI who present in acute care, screening must take place "within a very small window of time" [46]. Many studies included in this review did not report abstinence rates among their samples, and this was consequently another area from which we are not able garner much from the currently available research.

It is important to emphasise that the mechanisms underlying the development of ARCI remain unknown but are almost certainly multifactorial. Alcohol-related mechanisms including vitamin $B_1$ deficiency, genetic predisposition and co-occurring non-alcohol-related factors such as head injury, vascular dementia and age-related changes are all thought to contribute to the aetiology of ARCI [12]. Until we have a greater mechanistic understanding of ARCI, it is highly unlikely that a reproducible single set of diagnostic criteria will be developed. However, in the meantime, it is vital that consensus is sought, and standardised criteria agreed to aid clinical judgement, particularly as ARCI is often partially reversible. The ongoing lack of a consensus on standardised criteria has resulted in under recording and reporting of ARCI, with deleterious consequences for patients, families, health, and social care services. Ongoing evaluation of the practical application of new or existing ARCI criteria, either through formal research or routine monitoring, will be needed to confirm they have utility in clinical practice and that healthcare professionals do not face barriers to their use. The role of cognitive screening tools in supporting a suspected ARCI diagnosis requires greater attention, and in line with Ko et al., [45], we would caution against an over reliance on cut-off scores. Confounding factors may complicate assessment [3], and patients with suspected ARCI may commonly present with physical and psychiatric comorbidities [8]. Another important consideration is the 'double stigma' experienced by patients with ARCI within healthcare settings [6, 47]. Negative attitudes among healthcare professionals towards people with problems with alcohol or other substances are known to undermine access to diagnosis, treatment, and successful health

outcomes [48, 49]. Poor compliance with treatment is erroneously perceived as "feckless" with very little recognition of this patient group's diminished ability to understand, remember, and therefore comply with clinical advice and treatment [21]. ARCI criteria must therefore be both practical for screening the large numbers of patients presenting in acute hospital and community settings, cognisant of the range of contributing factors that may form part of an ARCI patient's presentation (including comorbid diagnoses and ongoing alcohol and poly-substance use) and allow for comparisons of different interventions to be made. Until we are comparing patients who are 'like with like' in research studies, confusion will hamper progress.

### Limitations of the scoping review process

We prospectively developed a protocol for this scoping review, but it was not prospectively registered as it was not eligible for registration with PROSPERO—the international prospective register of systematic reviews. Prospective registration of systematic reviews and scoping reviews aims to reduce bias in the conduct and reporting of research, and we therefore acknowledge this is a key limitation in our scoping review process, Further, as previously noted, the inclusion criteria for this review were broadened out during the review process and a wide range of study designs and approaches are subsequently included in the review. This limits the extent to which the evidence can be considered as a whole, and its overall coherence discussed. There is a lack of consistency across the studies identified in terms of the approach to exploring the screening, diagnosis, or assessment of patients with suspected ARCI, which reflects a lack of consensus on definitions of ARCI. Diagnostic criteria relevant for ARCI have not been extensively tested and the reasons underlying the adaptions of the existing criteria for ARD were not well described. It is possible that relevant studies were missed for inclusion in the review. However, as the review team had an expectation that the identification of relevant studies would be complicated by the wide use of terminology in the field, systematic literature searches were carried out across two phases. Further the review team included people with expertise both in systematic review methodologies and primary research and clinical practice with the patient group.

Further, the review of studies relevant to the screening, diagnosis, or assessment of cognitive impairment among patients with AUD excluded studies with heterogenous samples of patients with substance use disorder. Focusing on homogenous samples of AUD patients may limit the generalisability of this aspect of the review in terms of how representative the findings are of real-world clinical practice.

## Conclusion

The findings of this review confirm the scarcity of evidence available on the screening, diagnosis or assessment of patients with suspected ARCI. The lack of evidence is an important barrier to the development of clear guidelines for diagnosing ARCI, which would ultimately improve the real-world management and treatment of patients with ARCI. Progress is therefore urgently needed in the development of a consensus in defining the diagnostic criteria for ARCI.

## Supporting information

**S1 Protocol. Final protocol (October 2019) [revised December 2021].**
(PDF)

**S1 Table. Search strategy example for Ovid MEDLINE(R).**
(PDF)

**S2 Table. Critical appraisal of studies about the accuracy, validity and/or reliability of a cognitive assessment tool.**
(PDF)

**S1 Appendix. Complete data extraction tables.**
(PDF)

## Acknowledgments

We are grateful to Dr Amanda Atkinson, Public Health Institute, Liverpool John Moores University who contributed to study screening in the initial phase of the project, and Professor Harry Sumnall, also of Public Health Institute, Liverpool John Moores University who provided advice on the development of the systematic scoping review methods.

## Author Contributions

**Conceptualization:** Lynn Owens, Ian Gilmore, Paul Richardson.

**Formal analysis:** Andrew Thompson.

**Funding acquisition:** Lynn Owens, Ian Gilmore, Paul Richardson.

**Methodology:** Lisa Jones.

**Validation:** Andrew Thompson.

**Writing – original draft:** Lisa Jones, Lynn Owens, Andrew Thompson, Ian Gilmore, Paul Richardson.

**Writing – review & editing:** Lisa Jones, Lynn Owens, Andrew Thompson, Ian Gilmore, Paul Richardson.

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
