## [Decision Letter · Decision Letter 0]

10 Mar 2022

PONE-D-22-01818Development of diagnostic criteria for differential diagnosis of alcohol-related brain injury (ARBI) among heavy drinkers: a systematic scoping reviewPLOS ONE

Dear Dr. Jones,

Thank you for submitting your manuscript to PLOS ONE. After careful consideration, we feel that it has merit but does not fully meet PLOS ONE’s publication criteria as it currently stands. Therefore, we invite you to submit a revised version of the manuscript that addresses the points raised during the review process.

We look forward to receiving your revised manuscript.

Kind regards,

A/Prof Victoria Manning

Academic Editor

PLOS ONE

Journal Requirements:

(We are grateful to Dr Amanda Atkinson, Public Health Institute, Liverpool John Moores University who contributed to study screening in the initial phase of the project, and Professor Harry Sumnall, also of Public Health Institute, Liverpool John Moores University who provided advice on the development of the systematic scoping review methods. This study was funding by charitable funds from the Gastroenterology Fund, (formerly) Royal Liverpool and Broadgreen University Hospitals NHS Trust. The funders had no role in study design, data collection and analysis, decision to publish, or preparation of the manuscript.)

(No. The funders had no role in study design, data collection and analysis, decision to publish, or preparation of the manuscript.)

Additional Editor Comments:

Thank you for submitting your paper. I am very grateful to the two expert reviewers for their thorough read of your manuscript and for their insightful suggestions. Like the reviewers, I feel the paper has its strengths and addresses an important issue, but feel it would benefit from a much stronger rationale –by explaining some of issues, limitations with the existing criteria. At the moment it lacks a clearly defined purpose and reads more of a summary of what others have done and where they overlap. By providing some context in terms of the complexity, difficulties we face in defining/diagnosing alcohol-related cognitive impairment its contribution to the literature will become a lot clearer.

When revising your manuscript, I would encourage you to refer the recent systematic review by Ko et al (2021) and to look carefully at the conditions under which cognitive impairment screening tools can be used.

Other minor issues

Make it clear that the statement below refers to people with Korsakoff’s, particularly since this is a criticism of the recent, similar Heirene et al. (17) review

With treatment, Smith and Hillman (13) estimated that full recovery can be achieved in approximately 25% of patients, and among the remainder, 50% can achieve a partial or minor recovery

The paper would benefit from starting clearer objectives. Currently aim 2 (on page 4) seems rather weak, “the identification of further work in identifying the best tools”, surely this would be a by-product of addressing aim 1 “summarising the evidence of existing tools/criteria and any validations of those”.

As noted by one of the reviewers, the fact that a protocol was developed in advance (but not registered) is a major limitation and should be acknowledged.

Finally with regards to the title - perhaps it needs a word like "Towards developing" or "Informing the development" as it appears rather overstated in its current form, relative to what the paper describes.

I am confident you will find the reviewers comments helpful in the revision of the manuscript

Reviewers' comments:

Reviewer's Responses to Questions

**Comments to the Author**

1. Is the manuscript technically sound, and do the data support the conclusions?

Reviewer #1: Yes

Reviewer #2: Yes

2. Has the statistical analysis been performed appropriately and rigorously? 

Reviewer #1: N/A

Reviewer #2: N/A

3. Have the authors made all data underlying the findings in their manuscript fully available?

Reviewer #1: Yes

Reviewer #2: Yes

4. Is the manuscript presented in an intelligible fashion and written in standard English?

Reviewer #1: Yes

Reviewer #2: Yes

5. Review Comments to the Author

Reviewer #1: Thank you for the opportunity to review this manuscript entitled Development of diagnostic criteria for differential diagnosis of alcohol-related brain injury among heavy drinkers: a systematic scoping review. This manuscript is a timely and useful review highlighting an important issue in the addiction field regarding the lack of criteria for diagnosing alcohol related brain injury.

Overall the manuscript is of an appropriate quality and followed a systematic scoping review methodology to extract key themes from the included literature. Prior to recommendation for publication, however, I would like to see the following comments addressed.

General comments

Overall I missed a more in-depth critique of the currently available criteria. For instance, specifically what are the current issues or practical concerns with applying the available criteria? I think inclusion of this within the introduction and discussion would be helpful for readers in highlighting the specific issues experienced by clinicians along with some potential avenues for improvement that future studies could incorporate or explore. This would help strengthen the discussion which has only a very small section devoted to this issue before moving onto screening tools.

Introduction

In the introduction/methods, it might be helpful for Table 1 to be expanded to include all available formal diagnostic classification systems for ARBI/ARD. This would help summarise the lay of the land for readers and provide a better context for the review.

Further to this, I think the manuscript could benefit from some additional commentary about the DSM and ICD diagnostic classification systems including any strengths and identified limitations.

Methods

The systematic scoping review procedure was well executed and described.

Please provide a stronger justification for excluding studies that solely used DSM/ICD classifications and a statement regarding the number of studies that were excluded on this basis.

The exclusion criteria regarding mixed samples of drug and alcohol users is a reasonable one, however, this should be acknowledged as a general limitation given the clinical reality that polysubstance use is often the norm in AOD settings rather than the exception and research studies often struggle to reflect this.

Page 6 paragraph 3: Typo. Diagnosis should be diagnose.

Results

Table 3 – Note that the MoCA now requires training certification to administer. Note also that some assessment measures have user restrictions that prevent unqualified individuals being able to purchase or administer the measure. For instance, the RBANS and NIS have a User Level B (Psychologist, Speech Pathologist, Occupational Therapist, Special Education Teacher, Human Resources Professional, Psychiatrist, Paediatrician). Check with each test publisher to determine the User level.

In highlighting these screening tools I would recommend a note of caution that they ought to be administered and interpreted by an appropriately qualified health professional with experience in the field of cognition. For instance, I note that the TEDCA personnel use is “unclear” given the lack of formal guidance by a test publisher or equivalent, however, it is clear that the measure contains several neuropsychological assessment tools that are generally recommended for use by clinical neuropsychologists and I would not consider them to fall into the category of “easy brief administration by non-specialised healthcare personal in the clinical and research field”, as was concluded by the TEDCA study authors!

A good description of the concerns associated with the use of screening tools in the AOD sector is in the second paragraph of the clinical recommendations section of the following paper: doi:10.1017/S135561772100103X.

This paper also includes an extensive review of screening tools utilised in the AOD sector and would be worth cross checking and discussing where appropriate. For instance, their conclusions are quite similar in that very few studies have validated the use of screening tools in AOD populations with the MoCA being the most commonly used.

Table 3 includes abbreviations that were not described in the table notes: ROCFT, TMT.

In the section describing the findings of studies using screening measures (page 16-17), was abstinence discussed in any of these studies? If this data is available in the included studies it may be worth including as a column in Tables 4 & 5 to aid interpretation of the findings.

Did any of the included studies present information regarding comorbid diagnoses that could impact on the provision of a diagnosis of ARBI?

Discussion

The discussion may benefit from commentary on the ICD-11 now being in effect given there has been a change in the diagnostic criteria for alcohol related dementia compared with the ICD-10. Highlighting this and discussing any avenues for further development could be valuable for readers.

Within paragraph three could the authors provide some commentary on any observations regarding the presence of psychiatric comorbidity or other factors observed in the included studies that may impact the differential diagnosis of ARBI in this cohort?

Reviewer #2: You Plos One

Feb 2022

OVERALL COMMENTS

Thank you for the opportunity to review this interesting manuscript. This is a very important systematic review that highlights a real gap in the literature regarding this condition. It notes a paradox: we (myself included) consistently call for valid diagnostic criteria for ARBD/ARBI/ARNI but have failed to invest resources into developing them through rigorous empirical work. There are a few areas where I think the introduction/ discussion could be expanded to better account for the complexity involved in defining and diagnosing this condition, but these can be easily addressed in a revision of the paper.

Below I provide specific comments on each of the individual sections within the manuscripts.

INTRODUCTION

The authors choose to use the term “alcohol related brain injury”. This term is not widely used outside of very small circles in the UK and is semantically confusing/ambiguous — “injury” suggests the person has sustained a physical injury, and the term overall insinuates that somebody has experienced a brain injury that occurred whilst intoxicated, which is certainly not what the authors are referring to. I strongly suggest the authors consider alternative terms if they wish to attract international attention with this work, some of which I discuss below:

• Alcohol related brain damage - this phrase has been used in several studies published by myself and others in the UK over the last few years and is widely recognised by many clinicians in the UK, although it is admittedly not widely used outside of the UK. Further, some clinicians dislike the use of terms like “damage” and “injury” as they imply permanent physical damage. See the following paper (please note, I am referring you to this paper that I authored as I think it is relevant to this point and several others in your paper, but I do not expect you to cite the paper or discuss it in any way if you do not feel it is suitable/relevant to your work):

Heirene, R. M., John, B., O’Hanrahan, M., Angelakis, I. & Roderique-Davies, G. (2021). Professional Perspectives on Supporting Those with Alcohol-Related Neurocognitive Disorders: Challenges & Effective Treatment. Alcoholism Treatment Quarterly, 39(3), 1–27. https://doi.org/10.1080/07347324.2021.1898294

Full text access: https://robheirene.netlify.app/publication/heirene_et_al_2021_perspectives_on_arbd_treatment/Heirene_et_al_%282021%29_Perspectives_on_ARBD_treatment.pdf

Alcohol-related cognitive impairment - this phrase is unambiguous, uncontroversial, and widely accepted worldwide

• alcohol-related neurocognitive impairment - again, this phrase is unambiguous, uncontroversial, and widely accepted worldwide. This term is also more semantically accurate and simply cognitive impairment”.

It is good that the authors note the other diagnostic terms used in the confusion this brings.

Third paragraph: the Smith and Hillman study referred to here only discussed recovery rates in those with Korsakoff’s syndrome. The authors appear to exclude this manifestation from ”ARBI” in their introduction, no? If they include this syndrome under their definition of ARBI then this needs to be clarified in the first paragraph or two.

Final paragraph: further to my first comment, the authors state that criteria for ARD have been adapted for a probable diagnosis of “ARBI” but Wilson actually used the term “alcohol related brain damage” in the article cited.

Given the overall purpose of this review, the authors need to better clarify what they mean by “ARBI” at the start of the manuscript. What is and isn’t included under this definition? Can you detailed description of the symptomology be provided, whilst acknowledging heterogeneity in presentation and noting this as key? It needs to be absolutely clear what “condition” is been referred to before we move on to determining how best to screen for it. Later in the article, the authors refer to a variety of different study populations as having “ARBI” including those in Brown et al. and Wester et al., but these are described differently by their respective authors. Perhaps the authors could discuss some of these studies and the samples studied in the introduction as examples of populations with ARBI according to their definition?

METHODS

Why was the protocol not registered in advanced if it was developed prior to conducting the review? This seems very strange and needs clarifying.

INCLUSION CRITERIA

the objectives stated here differ slightly from those stated at the end of the introduction? There is some inconsistency here.

Were reviews included? Please clarify

Was there a language restriction? Please clarify

The authors state they excluded samples “where they included mixed samples of drug and alcohol users.” Did this include samples will use both alcohol and drugs simultaneously? Please clarify. Polysubstance misuse is common and among this group.

SEARCHING FOR AND SELECTING RELEVANT STUDIES

This process of searching for and selecting articles seems scientifically valid and thorough. It is great that the authors used three reviewers to independently screen for potential studies at the full text level. At the title and abstract screening level, did the authors record the level of consistency between the two reviewers when screening the initial 20% of articles? It would be good to have some indication of interrater reliability.

DATA CHARTING PROCESS AND QUALITY ASSESSMENT

DATA SYNTHESIS

I have no other concerns or comments regarding the other subsections of the methods section—everything is clearly explained.

RESULTS

in the first paragraph of this section, the authors state that “20 records were kept and comprised the final set of included studies in the review”, but figure 1 shows only 19 studies were kept. Please clarify which is correct.

The authors state that only one study (32) included patients with a diagnosis of ARBI. However, our study (19; see below) included participants with a diagnosis of alcohol related brain damage, which the authors appear to include in the definition of ARBI. Please clarify whether this is because you only included studies that diagnose participants according to the ARD criteria.

19: Brown P, Heirene RM, John B, Evans JJ. Applicability of the ACE-III and RBANS cognitive tests for the detection of Alcohol-Related Brain Damage. Frontiers in psychology. 2019;10:2636.

Page 14: can the authors please provide a brief narrative summary of the results from the quality assessment process? I think it’s fine to include the full table of outcomes in supplemental information, but the readers need to have some indication of what the outcomes were here for ease of interpretation. A simple sentence or two stating what percentage of studies met each of the criteria in the assessment would suffice.

DISCUSSION

First paragraph: the authors provide a clear and succinct summary of the findings here and make some important inferences regarding the lack of research in this area despite calls for validation of diagnostic criteria.

Third paragraph: the varied mechanisms underlying ARBI is an important point here. I think we have a good enough understanding of the mechanisms involved to know that they are multiple and varied. As a corollary, we shouldn’t expect there to be strict diagnostic/ screening criteria that apply to every case of ARBI/ARBD/ARNI. I think the authors need to make more of a point of this and, although we need to better test existing and new criteria and cut-off scores for these, we should recognise that there will always be some degree of clinical judgement involved in making this diagnosis. It seemed like the complexity diagnosis and heterogeneity in presentation is not fully appreciated here and could be expanded upon.

The conclusions derived are appropriate and important.

FINAL COMMENTS

I hope the authors find the above comments useful in revising their manuscripts for publication.

Sincerely,

Rob Heirene (I sign my reviews)

6. PLOS authors have the option to publish the peer review history of their article (what does this mean?). If published, this will include your full peer review and any attached files.

Reviewer #1: **Yes: **Dr James R. Gooden

Reviewer #2: **Yes: **Robert Heirene

---

## [Author Response · Author response to Decision Letter 0]

12 Jul 2022

EDITOR COMMENTS (E): Thank you for submitting your paper. I am very grateful to the two expert reviewers for their thorough read of your manuscript and for their insightful suggestions. Like the reviewers, I feel the paper has its strengths and addresses an important issue, but feel it would benefit from a much stronger rationale – by explaining some of issues, limitations with the existing criteria. At the moment it lacks a clearly defined purpose and reads more of a summary of what others have done and where they overlap. By providing some context in terms of the complexity, difficulties we face in defining/diagnosing alcohol-related cognitive impairment its contribution to the literature will become a lot clearer. When revising your manuscript, I would encourage you to refer the recent systematic review by Ko et al (2021) and to look carefully at the conditions under which cognitive impairment screening tools can be used.

RESPONSE: Thank you for you positive and constructive comments on the paper. We have reorganised and added to the Introduction to make the context and purpose of the paper clearer to the reader.

EC: Make it clear that the statement below refers to people with Korsakoff’s, particularly since this is a criticism of the recent, similar Heirene et al. (17) review

With treatment, Smith and Hillman (13) estimated that full recovery can be achieved in approximately 25% of patients, and among the remainder, 50% can achieve a partial or minor recovery

RESPONSE: This has been clarified in the text as follows (pg 4, 1st para): "Evidence from the broader spectrum of conditions associated with cognitive impairment suggests that recognition and a diagnosis of ARCI would significantly improve the long-term prognosis for patients. Among patients with Korsakoff’s Syndrome (KS), Smith and Hillman..."

EC: The paper would benefit from starting clearer objectives. Currently aim 2 (on page 4) seems rather weak, “the identification of further work in identifying the best tools”, surely this would be a by-product of addressing aim 1 “summarising the evidence of existing tools/criteria and any validations of those”. 

RESPONSE: We have deleted the following part of the sentence to ensure that the objectives of the paper/review are clearer: “the purpose of supporting further work to explore which of the available tools and/or operational criteria best facilitate the rapid identification of patients with suspected ARBI”.

EC: As noted by one of the reviewers, the fact that a protocol was developed in advance (but not registered) is a major limitation and should be acknowledged. 

RESPONSE:At the time of developing the protocol, we were aware of PROSPERO, the international prospective register of systematic reviews. We planned to register the protocol with PROSPERO but found that it does not accept scoping reviews. We were not aware of other free options for prospective registration at the time of developing the protocol. We now understand that paid options are available and/or the protocols can be published on free open science platforms. We have acknowledged that the protocol was not prospectively available in the limitations.

EC: Finally with regards to the title - perhaps it needs a word like "Towards developing" or "Informing the development" as it appears rather overstated in its current form, relative to what the paper describes. 

RESPONSE: ‘Informing’ added

EC: I am confident you will find the reviewers comments helpful in the revision of the manuscript.

RESPONSE: We have. Thank you.

REVIEWER #1 COMMENTS 

Reviewer #1: Thank you for the opportunity to review this manuscript entitled Development of diagnostic criteria for differential diagnosis of alcohol-related brain injury among heavy drinkers: a systematic scoping review. This manuscript is a timely and useful review highlighting an important issue in the addiction field regarding the lack of criteria for diagnosing alcohol related brain injury.

RESPONSE: Thank you for your comments. No response needed.

R1:Overall the manuscript is of an appropriate quality and followed a systematic scoping review methodology to extract key themes from the included literature. Prior to recommendation for publication, however, I would like to see the following comments addressed.

RESPONSE: Please see below for how individual comments have been addressed.

GENERAL COMMENTS 

R1: Overall I missed a more in-depth critique of the currently available criteria. For instance, specifically what are the current issues or practical concerns with applying the available criteria? I think inclusion of this within the introduction and discussion would be helpful for readers in highlighting the specific issues experienced by clinicians along with some potential avenues for improvement that future studies could incorporate or explore. This would help strengthen the discussion which has only a very small section devoted to this issue before moving onto screening tools.

RESPONSE: Please see below under Introduction for how we have responded to the comment about the need for more information about the currently available criteria.

INTRODUCTION 

R1: In the introduction/methods, it might be helpful for Table 1 to be expanded to include all available formal diagnostic classification systems for ARBI/ARD. This would help summarise the lay of the land for readers and provide a better context for the review. Further to this, I think the manuscript could benefit from some additional commentary about the DSM and ICD diagnostic classification systems including any strengths and identified limitations.

RESPONSE: We have expanded the section on ICD and DSM systems in the second paragraph to explain why they are not used in clinical practice to diagnosis ARCI. One the main issues, which we have sought to make clearer, is that there is a shortage of rigorous empirical research done in clinical practice to establish the validity/usefulness of the new and existing criteria. 

METHODS 

R1: The systematic scoping review procedure was well executed and described.

RESPONSE: Thank you for your comments. No response needed.

R1: Please provide a stronger justification for excluding studies that solely used DSM/ICD classifications and a statement regarding the number of studies that were excluded on this basis.

RESPONSE: We have clarified in the text that this exclusion was specific to ICD-10/11 and DSM-IV as neither version provides unifying criteria for ARCI. We dropped the diagnosis terms in our revised strategy and in practice no exclusions were made because of this criterion.

R1: The exclusion criteria regarding mixed samples of drug and alcohol users is a reasonable one, however, this should be acknowledged as a general limitation given the clinical reality that polysubstance use is often the norm in AOD settings rather than the exception and research studies often struggle to reflect this.

RESPONSE: We have added a paragraph about the exclusion of studies at the start of the results section. We acknowledge that polysubstance use is common among the patient group and did not exclude samples reporting polysubstance use. As we did not include explicit inclusion criteria in relation to other substance use disorders, we have added the following text to clarify what types of studies were excluded (pg 7; 4th para):"Twelve studies were considered ‘near misses’ and judged to have narrowly failed to meet the inclusion criteria in respect of the review’s secondary objective as the samples included patients with diagnoses of dependence on substances other than alcohol. We agreed post hoc to exclude these studies where the sample included patients with a primary diagnosis (based on DSM or other recognised criteria) of abuse or dependence on a substance other than alcohol." We have acknowledged that this may be a limitation in the Discussion (pg 20, 2nd para of Limitations).

R1: Page 6 paragraph 3: Typo. Diagnosis should be diagnose.

RESPONSE: This typo has been corrected.

RESULTS 

R1: Table 3 – Note that the MoCA now requires training certification to administer. Note also that some assessment measures have user restrictions that prevent unqualified individuals being able to purchase or administer the measure. For instance, the RBANS and NIS have a User Level B (Psychologist, Speech Pathologist, Occupational Therapist, Special Education Teacher, Human Resources Professional, Psychiatrist, Paediatrician). Check with each test publisher to determine the User level. In highlighting these screening tools I would recommend a note of caution that they ought to be administered and interpreted by an appropriately qualified health professional with experience in the field of cognition. For instance, I note that the TEDCA personnel use is “unclear” given the lack of formal guidance by a test publisher or equivalent, however, it is clear that the measure contains several neuropsychological assessment tools that are generally recommended for use by clinical neuropsychologists and I would not consider them to fall into the category of “easy brief administration by non-specialised healthcare personal in the clinical and research field”, as was concluded by the TEDCA study authors! A good description of the concerns associated with the use of screening tools in the AOD sector is in the second paragraph of the clinical recommendations section of the following paper: doi:10.1017/S135561772100103X. This paper also includes an extensive review of screening tools utilised in the AOD sector and would be worth cross checking and discussing where appropriate. For instance, their conclusions are quite similar in that very few studies have validated the use of screening tools in AOD populations with the MoCA being the most commonly used. 

RESPONSE: Thank you for the orientation to the Ko et al., paper this has usefully informed our revisions. With respect to clinical utility, we have removed the column that refers to Personnel in Table 3, as on reflection this was not helpful with respect to the purposes of the paper and based on that (overall) this information was incomplete and partial across the different tools.

R1: Table 3 includes abbreviations that were not described in the table notes: ROCFT, TMT. 

RESPONSE: The list of abbreviations has been checked and the missing abbreviations added.

R1: In the section describing the findings of studies using screening measures (page 16-17), was abstinence discussed in any of these studies? If this data is available in the included studies it may be worth including as a column in Tables 4 & 5 to aid interpretation of the findings.

RESPONSE: Information about abstinence was extracted and is presented in the data extraction tables and quality assessment information in the supporting information. We have added a paragraph about the exclusion of studies at the start of the results section (based on Reviewer #2 comment) and included additional information here about abstinence. Brief information has also been added to Tables 4 and 5.

R1: Did any of the included studies present information regarding comorbid diagnoses that could impact on the provision of a diagnosis of ARBI?

RESPONSE: Information about sample exclusions was extracted and is presented in the data extraction tables and quality assessment information in the supporting information. We have added a paragraph about the exclusion of studies at the start of the results section (based on Reviewer #2 comment) and included additional information here about exclusions based on co-morbidity.

DISCUSSION 

R1: The discussion may benefit from commentary on the ICD-11 now being in effect given there has been a change in the diagnostic criteria for alcohol related dementia compared with the ICD-10. Highlighting this and discussing any avenues for further development could be valuable for readers. 

RESPONSE: We have added information to the Introduction about the DSM and ICD classification systems. Although we accept the comment and agree that is could be beneficial to add further commentary in the Discussion we are have chosen not to address this comment as it would add significantly to the word count. 

R1: Within paragraph three could the authors provide some commentary on any observations regarding the presence of psychiatric comorbidity or other factors observed in the included studies that may impact the differential diagnosis of ARBI in this cohort? 

RESPONSE: We have added to paragraph three, noting that confounding factors may complicate assessment of patients with suspected ARCI.

REVIEWER #2 COMMENTS

OVERALL COMMENTS 

R2: Thank you for the opportunity to review this interesting manuscript. This is a very important systematic review that highlights a real gap in the literature regarding this condition. It notes a paradox: we (myself included) consistently call for valid diagnostic criteria for ARBD/ARBI/ARNI but have failed to invest resources into developing them through rigorous empirical work. There are a few areas where I think the introduction/ discussion could be expanded to better account for the complexity involved in defining and diagnosing this condition, but these can be easily addressed in a revision of the paper. RESPONSE: Thank you for your comments. Please see below for our responses to each comment.

INTRODUCTION 

R2: The authors choose to use the term “alcohol related brain injury”. This term is not widely used outside of very small circles in the UK and is semantically confusing/ambiguous — “injury” suggests the person has sustained a physical injury, and the term overall insinuates that somebody has experienced a brain injury that occurred whilst intoxicated, which is certainly not what the authors are referring to. I strongly suggest the authors consider alternative terms if they wish to attract international attention with this work, some of which I discuss below:

• Alcohol related brain damage - this phrase has been used in several studies published by myself and others in the UK over the last few years and is widely recognised by many clinicians in the UK, although it is admittedly not widely used outside of the UK. Further, some clinicians dislike the use of terms like “damage” and “injury” as they imply permanent physical damage. See the following paper (please note, I am referring you to this paper that I authored as I think it is relevant to this point and several others in your paper, but I do not expect you to cite the paper or discuss it in any way if you do not feel it is suitable/relevant to your work):

Heirene, R. M., John, B., O’Hanrahan, M., Angelakis, I. & Roderique-Davies, G. (2021). Professional Perspectives on Supporting Those with Alcohol-Related Neurocognitive Disorders: Challenges & Effective Treatment. Alcoholism Treatment Quarterly, 39(3), 1–27. https://doi.org/10.1080/07347324.2021.1898294

Full text access: https://robheirene.netlify.app/publication/heirene_et_al_2021_perspectives_on_arbd_treatment/Heirene_et_al_%282021%29_Perspectives_on_ARBD_treatment.pdf

Alcohol-related cognitive impairment - this phrase is unambiguous, uncontroversial, and widely accepted worldwide

Alcohol-related neurocognitive impairment - again, this phrase is unambiguous, uncontroversial, and widely accepted worldwide. This term is also more semantically accurate and simply cognitive impairment”. It is good that the authors note the other diagnostic terms used in the confusion this brings.

RESPONSE: Thank you for providing your considerations on this issue. The authors have discussed the terminology used and we have agreed with the suggestion to change the overarching term used to ‘Alcohol Related Cognitive Impairment’. The introduction has been reorganised to better clarify the sources of variation and how they have been used by clinicians and researchers.

Please see our responses to the next set of comments.

R2: Third paragraph: the Smith and Hillman study referred to here only discussed recovery rates in those with Korsakoff’s syndrome. The authors appear to exclude this manifestation from ”ARBI” in their introduction, no? If they include this syndrome under their definition of ARBI then this needs to be clarified in the first paragraph or two.

RESPONSE: This has been clarified in the text as follows (pg 4, 1st para): "Evidence from the broader spectrum of conditions associated with cognitive impairment suggests that recognition and a diagnosis of ARCI would significantly improve the long-term prognosis for patients. Among patients with Korsakoff’s Syndrome (KS), Smith and Hillman..."

R2: Final paragraph: further to my first comment, the authors state that criteria for ARD have been adapted for a probable diagnosis of “ARBI” but Wilson actually used the term “alcohol related brain damage” in the article cited.

RESPONSE: We have clarified this by changing to the phrasing used in the original paper: ‘probable diagnosis of alcohol-related brain damage’.

R2: Given the overall purpose of this review, the authors need to better clarify what they mean by “ARBI” at the start of the manuscript. What is and isn’t included under this definition? Can you detailed description of the symptomology be provided, whilst acknowledging heterogeneity in presentation and noting this as key? It needs to be absolutely clear what “condition” is been referred to before we move on to determining how best to screen for it. Later in the article, the authors refer to a variety of different study populations as having “ARBI” including those in Brown et al. and Wester et al., but these are described differently by their respective authors. Perhaps the authors could discuss some of these studies and the samples studied in the introduction as examples of populations with ARBI according to their definition? 

RESPONSE: As noted above, we have reorganised the opening paragraphs of the Introduction. This now establishes that alongside the use of variant terms, that as umbrella terms, they have also been used to both encompass and exclude WKS. We have added the following text as clarification about the focus of the scoping review (pg 3, end of 1st para): "For the purposes of this scoping review, we have used the term Alcohol-Related Cognitive Impairment (ARCI) and applied it in its broadest sense. We have included articles following either application of the terminology, for example, by recognising that ARD and ARCI may be used to refer to the same suspected condition, and that umbrella terms such as ARBD may either encompass or exclude WKS."

METHODS 

R2: Why was the protocol not registered in advanced if it was developed prior to conducting the review? This seems very strange and needs clarifying.

RESPONSE: At the time of developing the protocol, we were aware of PROSPERO, the international prospective register of systematic reviews. We planned to register the protocol with PROSPERO but it does not accept scoping reviews. We were not aware of other free options for prospective registration at the time of developing the protocol. We now understand that paid options were available or free open science platforms.

INCLUSION CRITERIA 

R2: The objectives stated here differ slightly from those stated at the end of the introduction? There is some inconsistency here. 

RESPONSE: We have checked the text for consistency between the end of the Introduction and the Inclusion criteria. We have amended the text to make it clearer that there were different selection criteria applied in respect of the two objectives for the review.

R2: Were reviews included? Please clarify 

RESPONSE: Reviews were not included and we have clarified that original, primary research were eligible.

R2: Was there a language restriction? Please clarify

RESPONSE: We have made it clear that only studies published in English were eligible.

R2: The authors state they excluded samples “where they included mixed samples of drug and alcohol users.” Did this include samples will use both alcohol and drugs simultaneously? Please clarify. Polysubstance misuse is common and among this group.

RESPONSE: We have added a paragraph about the exclusion of studies at the start of the results section. We acknowledge that polysubstance use is common among the patient group and did not exclude samples reporting polysubstance use. As we did not include explicit inclusion criteria in relation to other substance use disorders, we have added the following text to clarify what types of studies were excluded (pg 7; 4th para): "Twelve studies were considered ‘near misses’ and judged to have narrowly failed to meet the inclusion criteria in respect of the review’s secondary objective as the samples included patients with diagnoses of dependence on substances other than alcohol. We agreed post hoc to exclude these studies where the sample included patients with a primary diagnosis (based on DSM or other recognised criteria) of abuse or dependence on a substance other than alcohol." We have acknowledged that this may be a limitation in the Discussion (pg 20, 2nd para of Limitations).

SEARCHING FOR AND SELECTING RELEVANT STUDIES 

R2: This process of searching for and selecting articles seems scientifically valid and thorough. It is great that the authors used three reviewers to independently screen for potential studies at the full text level. At the title and abstract screening level, did the authors record the level of consistency between the two reviewers when screening the initial 20% of articles? It would be good to have some indication of interrater reliability. 

RESPONSE: Interrater reliability was not recorded. Because of the wide use of terminology in the field it was difficult on occasion to discern from the titles/abstracts whether a study contained relevant information or not, so a level of disagreement was expected in the dual screening process. The discussions about disagreements were therefore used to develop confidence in the selection process conducted by the single reviewer. The following text has been added (pg 6, 2nd para): "Interrater reliability was not recorded as a level of disagreement was anticipated based on the wide use of terminology in the field. Discrepancies in study selection were discussed and resolved and used to develop confidence in the screening of the remaining titles and abstracts which was carried out by a single reviewer (LJ)."

RESULTS 

R2: In the first paragraph of this section, the authors state that “20 records were kept and comprised the final set of included studies in the review”, but figure 1 shows only 19 studies were kept. Please clarify which is correct.

RESPONSE: The correct number of included studies is 19. The flowchart has been corrected for the errors that have crept into the final number of included studies and reasons for full-text exclusions.

R2: The authors state that only one study (32) included patients with a diagnosis of ARBI. However, our study (19; see below) included participants with a diagnosis of alcohol related brain damage, which the authors appear to include in the definition of ARBI. Please clarify whether this is because you only included studies that diagnose participants according to the ARD criteria.

RESPONSE: We have clarified in the text that this description applies to the six studies that were descriptive reports. Reference 19 is not referred to within this set of studies and is reported in the sentences preceding as examining the diagnostic criteria for ARBI (now referred to as ARCI).

R2: Page 14: can the authors please provide a brief narrative summary of the results from the quality assessment process? I think it’s fine to include the full table of outcomes in supplemental information, but the readers need to have some indication of what the outcomes were here for ease of interpretation. A simple sentence or two stating what percentage of studies met each of the criteria in the assessment would suffice.

RESPONSE: A short paragraph has been added about quality assessment (pg 14, 1st para): "All seven studies provided adequate descriptions of the included patient samples, but exclusion based on comorbid or confounding conditions was not clear or not reported in two studies (20, 27). Further, effect sizes were only reported in two of six studies where this was deemed appropriate (19, 31). As shown in Table 4, studies typically included small to moderate sample sizes. Four studies (19, 20, 28, 34) reported clearly that they had adjusted for the risk of type 1 error in their analyses."

DISCUSSION 

R2: First paragraph: the authors provide a clear and succinct summary of the findings here and make some important inferences regarding the lack of research in this area despite calls for validation of diagnostic criteria.

RESPONSE: Thank you for your comment. No response required.

R2: Third paragraph: the varied mechanisms underlying ARBI is an important point here. I think we have a good enough understanding of the mechanisms involved to know that they are multiple and varied. As a corollary, we shouldn’t expect there to be strict diagnostic/ screening criteria that apply to every case of ARBI/ARBD/ARNI. I think the authors need to make more of a point of this and, although we need to better test existing and new criteria and cut-off scores for these, we should recognise that there will always be some degree of clinical judgement involved in making this diagnosis. It seemed like the complexity diagnosis and heterogeneity in presentation is not fully appreciated here and could be expanded upon.

RESPONSE: We have added the additional considerations to the Discussion (pg 19, 2nd para): "However, evaluation of the practical application of new or existing ARCI criteria, either through formal research or routine monitoring, is needed to confirm that they have utility in clinical practice and that healthcare professionals do not face barriers to their use. Further, patients with ARCI report experiencing a ‘double stigma’ within healthcare settings (10, 40) and negative attitudes towards people with problems with alcohol or other substances among healthcare professionals are known to undermine access to diagnosis, treatment, and successful health outcomes (41, 42)."

R2: The conclusions derived are appropriate and important.

RESPONSE: Thank you for your comment. No response required.

---

## [Decision Letter · Decision Letter 1]

29 Aug 2022

PONE-D-22-01818R1Informing the development of diagnostic criteria for differential diagnosis of alcohol-related brain injury (ARBI) among heavy drinkers: a systematic scoping reviewPLOS ONE

Dear Dr Jones,

Thank you for submitting your manuscript to PLOS ONE. Thank you for carefully addressing the reviewers comments -the paper is much improved. We invite you to submit a revised version of the manuscript that addresses the final remaining point raised during the second review.

Please submit your revised manuscript by Oct 13 2022 11:59PM. If you will need more time than this to complete your revisions, please reply to this message or contact the journal office at plosone@plos.org. Please include the following items when submitting your revised manuscript:A rebuttal letter that responds to each point raised by the academic editor and reviewer(s). You should upload this letter as a separate file labeled 'Response to Reviewers'.A marked-up copy of your manuscript that highlights changes made to the original version. You should upload this as a separate file labeled 'Revised Manuscript with Track Changes'.An unmarked version of your revised paper without tracked changes. You should upload this as a separate file labeled 'Manuscript'.If applicable, we recommend that you deposit your laboratory protocols in protocols.io to enhance the reproducibility of your results. Protocols.io assigns your protocol its own identifier (DOI) so that it can be cited independently in the future. For instructions see: https://journals.plos.org/plosone/s/submission-guidelines#loc-laboratory-protocols. Additionally, PLOS ONE offers an option for publishing peer-reviewed Lab Protocol articles, which describe protocols hosted on protocols.io. Read more information on sharing protocols at https://plos.org/protocols?utm_medium=editorial-email&utm_source=authorletters&utm_campaign=protocols.

We look forward to receiving your revised manuscript.

Kind regards,

A/Prof Victoria Manning

Academic Editor

PLOS ONE

Journal Requirements:

Additional Editor Comments:

I agree with reviewer 1 that the paper would be strengthened with the addition of just a few sentences explaining how and why there were adaptations to the criteria.

Reviewers' comments:

Reviewer's Responses to Questions

**Comments to the Author**

1. If the authors have adequately addressed your comments raised in a previous round of review and you feel that this manuscript is now acceptable for publication, you may indicate that here to bypass the “Comments to the Author” section, enter your conflict of interest statement in the “Confidential to Editor” section, and submit your "Accept" recommendation.

Reviewer #1: (No Response)

Reviewer #2: All comments have been addressed

2. Is the manuscript technically sound, and do the data support the conclusions?

Reviewer #1: Yes

Reviewer #2: Yes

3. Has the statistical analysis been performed appropriately and rigorously? 

Reviewer #1: Yes

Reviewer #2: N/A

4. Have the authors made all data underlying the findings in their manuscript fully available?

Reviewer #1: Yes

Reviewer #2: Yes

5. Is the manuscript presented in an intelligible fashion and written in standard English?

Reviewer #1: Yes

Reviewer #2: Yes

6. Review Comments to the Author

Reviewer #1: Thank you for the opportunity to review this revised manuscript and I thank the authors for addressing the comments with some excellent additions to the manuscript.

I have some minor comments that the authors and editor may wish to consider prior to publication.

Discussion

I recognise this is a challenging point to address so it may not feasible, however, I still think the discussion could benefit from some elaboration on the available criteria and any emerging themes or issues from the included studies to help advance our thinking and maybe guide some next steps. For instance, a few of the included studies made adaptions to existing criteria, so what were these adaptions? Where they similar or not? Why did these authors feel the need to made such adaptions?

Another example worth raising is highlighting how disparate the criteria are vs rates of abstinence reported in the screening studies. Your description of the abstinence rates in the results is very illuminating because it shows many studies don’t adhere to the criteria, if indeed they even report abstinence rates in the first place! This is a worthy theme regarding any future criteria for clinicians and researchers to think about given the limitations in existing research and the practicalities of having patients maintain abstinence long enough to this meet criterion.

Minor:

As a reader there are a lot of abbreviations to keep track of and this risks adding to the already slightly confusing nature of all the different terms for alcohol related cognitive impairment. I would suggest considering a quick review of which abbreviations are absolutely necessary and which ones would read better in full. I realise this might add to the word count but I think for ease of reading it would be worthwhile.

Results (page 15): I was a little lost with the revised sentence following mention of Table S4 referring to “The tool from the review by Heirene…”. I was trying to understand “what tool” so maybe this paragraph just needs a slight restructure or framing for the reader to link the previous sentence with the next.

Reviewer #2: The authors have appropriately revised their manuscript according to the comments provided. I have no further concerns.

7. PLOS authors have the option to publish the peer review history of their article (what does this mean?). If published, this will include your full peer review and any attached files.

Reviewer #1: **Yes: **James R. Gooden

Reviewer #2: **Yes: **Robert Heirene

---

## [Author Response · Author response to Decision Letter 1]

9 Dec 2022

We have responded to the specific reviewer and editor comments in our Response to Reviewer attachment.

---

## [Editor Report · Decision Letter 2]

8 Jan 2023

Informing the development of diagnostic criteria for differential diagnosis of alcohol-related cognitive impairment (ARCI) among heavy drinkers: a systematic scoping review

We’re pleased to inform you that your manuscript has been judged scientifically suitable for publication and will be formally accepted for publication once it meets all outstanding technical requirements.

Kind regards,

Victoria Manning

Academic Editor

PLOS ONE

Additional Editor Comments (optional):

Thank you for your considered response to the issues and comments raised in the peer-review process. I think the review now provides a much more contextualised and nuanced account of what has been established to date and remaining knowledge gaps in relation to screening tools for alcohol-related cognitive impairment, and as a result it makes a useful contribution to the literature.
---

## [Editor Report · Acceptance letter]

16 Jan 2023

PONE-D-22-01818R2 

Informing the development of diagnostic criteria for differential diagnosis of alcohol-related cognitive impairment (ARCI) among heavy drinkers: a systematic scoping review 

Dear Dr. Jones:

I'm pleased to inform you that your manuscript has been deemed suitable for publication in PLOS ONE. Congratulations! Your manuscript is now with our production department. 

Kind regards, 

on behalf of

Dr. Victoria Manning 

Academic Editor

PLOS ONE